# Genome-Wide Identification of Calmodulin-Binding Protein 60 Gene Family and the Function of *GhCBP60B* in Cotton Growth and Development and Abiotic Stress Response

**DOI:** 10.3390/ijms25084349

**Published:** 2024-04-15

**Authors:** Kun Luo, Long Sha, Tengyu Li, Chenlei Wang, Xuan Zhao, Jingwen Pan, Shouhong Zhu, Yan Li, Wei Chen, Jinbo Yao, Junkang Rong, Yongshan Zhang

**Affiliations:** 1State Key Laboratory of Cotton Biology, Institute of Cotton Research, Chinese Academy of Agricultural Science, Anyang 455000, China; lk18871818259@163.com (K.L.); 15188358315@163.com (L.S.); tengyuli18@163.com (T.L.); wangchenlei18@163.com (C.W.); 13849259577@163.com (J.P.); zhushouhong@caas.cn (S.Z.); liyan06@caas.cn (Y.L.); chenwei01@caas.cn (W.C.); yaojinbo@caas.cn (J.Y.); 2The Key Laboratory for Quality Improvement of Agricultural Products of Zhejiang Province, Zhejiang Agricultural and Forestry University, Hangzhou 311300, China; 2021601022035@stu.zafu.edu.cn

**Keywords:** cotton, calmodulin-binding protein 60, growth and development, abiotic stress, *GhCBP60B*

## Abstract

The calmodulin-binding protein 60 (*CBP60*) family is a gene family unique to plants, and its members play a crucial role in plant defense responses to pathogens and growth and development. Considering that cotton is the primary source of natural cotton textile fiber, the functional study of its *CBP60* gene family members is critical. In this research, we successfully identified 162 *CBP60* members from the genomes of 21 species. Of these, 72 members were found in four cotton species, divided into four clades. To understand the function of *GhCBP60B* in cotton in depth, we conducted a detailed analysis of its sequence, structure, cis-acting elements, and expression patterns. Research results show that *GhCBP60B* is located in the nucleus and plays a crucial role in cotton growth and development and response to salt and drought stress. After using VIGS (virus-induced gene silencing) technology to conduct gene silencing experiments, we found that the plants silenced by *GhCBP60B* showed dwarf plants and shortened stem nodes, and the expression of related immune genes also changed. In further abiotic stress treatment experiments, we found that GhCBP60B-silenced plants were more sensitive to drought and salt stress, and their POD (peroxidase) activity was also significantly reduced. These results imply the vital role of *GhCBP60B* in cotton, especially in regulating plant responses to drought and salt stress. This study systematically analyzed *CBP60* gene family members through bioinformatics methods and explored in depth the biological function of *GhCBP60B* in cotton. These research results lay a solid foundation for the future use of the *GhCBP60B* gene to improve cotton plant type and its drought and salt resistance.

## 1. Introduction

Calcium is an essential nutrient for plant growth and development and has vital physiological functions [1]. Calcium ions can regulate plant seed germination rate, plant vegetative growth, cell polar growth, and cell wall biosynthesis in reproductive development. At the same time, when plants respond to adversity stress, they can alleviate low temperatures by activating protective enzyme systems and improving photosynthesis—high temperature, drought, high salt, diseases, insect pests, and other stresses that damage plants. Calcium also plays a regulatory role as a second messenger, coupling extracellular signals and intracellular physiological responses. It is essential to regulate plant growth and development and respond to biotic and abiotic stresses [2].

There are three large calcium ion-binding protein families in plants: calcineurin B-like (CBL) proteins, calcium-dependent protein kinase (CDPK) proteins, and calmodulin (CaM) and calmodulin-like (CML) proteins [3]. CaM is the central signal transduction molecule in the intracellular Ca^2+^ signaling pathway, mediating and regulating a series of physiological and biochemical reactions caused by Ca^2+^ [4]. The globular domain of the CaM protein binds Ca^2+^, and the exposed globular domain is called the CaM-binding structure domain (CBD), which interacts with calmodulin-binding proteins (CBPs) [5]. Among them, plants’ calmodulin-binding protein 60 (*CBP60*) family is a unique transcription factor family. It was discovered during identifying genes encoding calmodulin-binding proteins in the *Arabidopsis* genome [6]. Research shows that there are eight members of CBP60 protein family in Arabidopsis, including CBP60a g and systemic acquired resistance deficient 1 (*SARD1*); *SARD1* does not have a CaM-binding domain (CBD) that could bind to CAM [7]. 

The *CBP60* family plays an essential role in immunity and disease resistance. *CBP60g* and *SARD1* positively regulate immune responses in plants. It was found that *cbp60g/SARD1* mutant plants were highly susceptible to pathogenic bacteria [6]. Contrarily, *CBP60A* has been reported to regulate immune responses negatively in plants [8]. During plant development, the expression of immune genes in c*bp60a* increases, but these genes can be induced to wild-type levels after pathogen treatment. Recently, the *CBP60B* gene has been found to play a central role in the transcriptional regulation of plant immunity. It is constitutively expressed in uninjured plants and is closely related to the growth and development of plants. Moreover, the loss of *CBP60B* function can also lead to autoimmunity. PRs and SID2 are marker genes related to immunity and resistance in plants. When *CBP60B* is knocked out in Arabidopsis, the expression of these genes changes significantly [6]. *GhCBP60B* also plays a conservative and important role in cotton resistance. However, there are few studies related to the salt and drought stress of *CBP60* family in plants. Shen found that *SmCBP60A1* plays a positive regulatory role in the response to salt stress in eggplant, but the response of other members of *CBP60* gene family to stress in cotton remains to be studied [9].

As an important economic crop, cotton is the largest source of natural fibers to the textile industry [10]. Moreover, cotton, being a salt-tolerant crop, can help to make farmland more suitable for crop production [11]. More than 954.38 million hectares of saline–alkali lands are affecting crop yields and human food security adversely globally [12]. Although cotton is recognized as a pioneer crop in saline–alkali lands, its salt tolerance mechanism is not yet fully understood. The present study identified and analyzed 162 *CBP60* family genes from 21 species. The *GhCBP60s* genes and their phylogenetic relationships, structural diversity, staining distribution, and expression patterns under different stresses were identified. We also investigated the r role of *GhCBP60B* in response to salt and drought stress in cotton. Our findings showed that *GhCBP60B* plays a significant role in abiotic stress conditions.

## 2. Results

### 2.1. Identification and Phylogenetic Analysis of CBP60s

Gene family members of *CBP60* were identified to find functional characteristics and phylogenetic relationships with other species. Phylogenetic trees were constructed for 21 species (including four cotton species). We used the eight identified AtCBP60s protein sequences in *Arabidopsis thaliana* as query sequences. We used the BLAST tool to search the protein sequence libraries of other species, with the e value set to 1 × 10^−5^. We verified whether the calmodulin-binding protein-like (PF07887) family exists in the screened protein sequences through HMMER (http://www.ebi.ac.uk/Tools/hmmer/, accessed on 27 July 2023) and SMART (http:// smart.embl-heidelberg.de, accessed on 12 August 2023). A total of 162 *CBP60* gene family members were identified from 21 species (Figure 1A), among which no *CBP60* family members were placed in *Chlamydomonas reinhardtii* or *Volvox*. The *G. arboreum* (Ga), *G. raimondii* (Gr), *G. hirsutum* (Gh), and *G. barbadense* (Gb) family members in cotton are named based on their positional relationship to *CBP60s* in *Arabidopsis* in the evolutionary tree.

The distribution of 162 CBP60s proteins: 3 in *Amborella trichopoda* (Atr), 6 in *Ananas comosus* (Ac), 6 in *Cpapaya* (Cp), 8 in *Arabidopsis thaliana* (At), 19 in *Glycine max* (Gm), 3 in *Marchantia polymorpha* (Mp), 21 in *Oryza Sativa* (Os), 12 in *Populus trichocarpa* (Pt), 6 in *Physcomitrium patens* (Pp), 12 in *Solanum lycopersicum* (Sl), 14 in *Sorghum bicolor* (Sb), 2 in *Selaginella tamariscina* (Sm), 6 in *Theobroma cacao* (Tc), 9 in *Vitis vinifera* (Vv), 11 in *Vunguiculata* (Vu), *Zea mays* (Zm), and 12 in *G. raimondii* (Gr). The phylogenetic tree showed that the 162 protein sequences were divided into four branches (Ⅰ, Ⅱ, Ⅲ, Ⅳ) (Figure 1A). The four branches are named according to the AtCBP60s in *Arabidopsis thaliana*, which are CBP60B-F (Ⅰ), CBP60A (Ⅱ), SARD (Ⅲ), and CBP60G (Ⅳ). Among them, branch I appeared first in mosses and ferns. There are three and six members of mosses Mp and Pp, respectively. There are two CBP60 family members in *Selaginella tamariscina* (Sm), and both of these are present in the branch I, with no *CBP60* members found in earlier algae plants. Clade I is further differentiated into three immune regulatory factor subfamilies: CBP60a, CBP60g, and SARD1. The distribution of these members from each branch of dicotyledons and monocotyledons shows that subfamilies II, III, and IV evolved slower than the prototype group (I). 

Interestingly, *CBP60* genes in each subfamily were clustered in a monocot- or dicot-specific pattern. Compared with dicots, the proportion of monocots in *CBP60G* (branch IV) is significantly higher than that of dicots, indicating that branch IV has a faster expansion rate in monocots. A total of 12 members of the *CBP60* gene family were identified in diploid cotton Gr. Compared with dicotyledonous plants, e.g., cocoa (6) and grape (9), Gr expanded in branches I, II, and IV during the evolutionary process increase.

The genes of the *CBP60* family of cotton are named according to the phylogenetic tree and the homologs of *Arabidopsis thaliana*. “A” and “D” are added after the genes to distinguish the homologous genes on the A and D chromosomes (Figure 1B). In the four cotton varieties, a total of 72 *CBP60* genes were identified, 12 from the diploid cottons (*G. arboreum* and *G. raimondii*) and 24 from the tetraploid cottons (*G. hirsutum*, and *G. barbadense*). Among them, branch I contains the most *CBP60* members (6 *GaCBP60s*, 6 *GrCBP60s*, 12 *ChCBP60s*, 12 *GbCBP60s*, and 5 *AtCBP60s*), branch II contains 12 members (2*Ga*, 2*Gr*, 4*Ch*, 4*Gb*, 1*At*), branch III contains 12 members (2 *Ga*, 2 *Gr*, 4*Ch*, 4*Gb*, and 1*At*), and branch IV contains 12 members (2*Ga*, 2*Gr*, 4*Ch*, 4*Gb*, and 1*At*). Furthermore, these protein sequences were utilized to find physical properties, such as start–end locations, CDS length, protein length, protein molecular weight, pI, and subcellular localization, through online webservers, namely the ExPASy and Cell-PLoc online websites. The amino acid lengths of the 72 *CBP60* family members range from 458 (*GhSARD1-2D*) to 645 (*GhCBP60D-D*), and the protein isoelectric points range from 5.397 (*GhCBP60D-D*) to 8.659 (*GhSARD1-2D*) (Appendix A). *CBP60s* chromosome maps of four cotton species and phylogenetic relationship of the 162 identified CBP60 genes from 21 plant species (contains bootstrap values) is in Appendix A.

### 2.2. Chromosomal Localization of Cotton CBP60 Gene

To visualize the distribution of *CBP60s* on cotton chromosomes, genomes, and subgenomes, we mapped *CBP60s* chromosome maps of four cotton species (Appendix A). The *CBP60s* genes of Gh and Gb in tetraploid cotton are exactly double that of diploid Gr and Ga. The same number of *CBP60s* on every chromosome of tetraploid cotton, as well as on the diploid cotton, indicates that the *CBP60s* gene in cotton is relatively conservative. The 24 *CBP60s* genes are unevenly distributed on the 16 chromosomes of Gh (12 genes on each of the A and D genomes) and respond one-to-one at the positions on the chromosomes of the A and D genomes, without tandem duplication. *GhCBP60s* genome mapping results show that a total of 24 genes were mapped onto 16 chromosomes, in addition to the AD-1-A01, AD1-A02, AD1-A04, AD1-A07, AD1-A11, AD1-D01, AD1-D02, AD1-D04, AD1-D07, and AD1-D11 chromosomes (Appendix A). Apart from three *GhCBP60B* genes located on AD1-A08 and AD1-D08, there are two more *GhCBP60* genes that are present on each chromosome (AD1-A012, AD1-A13, AD1-D12, and AD1-D13), and only one *GhCBP60* is located on other chromosomes.

### 2.3. Phylogenetic Classification, Gene Structure, and Conserved Motif Analyses of GhCBP60

Genes’ structures, motifs, and domains are closely related to their functions. We analyzed the sequence structure, motifs, and conserved domains of the identified cotton *CBP60*. The phylogeny of upland cotton and *Arabidopsis CBP60* shows that the *CBP60* protein sequence of upland cotton is divided into four branches, which is consistent with the previous phylogenetic studies, namely *CBP60B-F* (I), *CBP60A* (II), *SARD* (III), and *CBP60G* (IV) four branches (Figure 2A). Among them, the gene structure analysis of four (Ⅰ to Ⅳ) branches shows that the number of *GhCBP60* introns ranges from 6 to 7, and the number of exons ranges from 7 to 8, which is highly conserved within the four branches (Figure 2C). MEME software (http://meme-suite.org/tools/meme, accessed on 3 October 2023) was used to identify the conserved motifs of the *CBP60* gene. A total of 10 conserved motifs were identified, named motif1~motif10 (Figure 2B). Within the same branch, the conserved motifs contain similar motifs. The overall structure of the cotton *CBP60* protein family is quite conserved. All conserved motifs, except for motifs 8 and 9, are present in every CBP60 homolog. This suggests that these motifs are crucial components of the GhCBP60 family. There are variations within different branches of *CBP60* homologs. *CBP60A* (II) has an additional section of motif 1 after the sequence compared to other branches, while *SARD* (III) does not have motifs 8 and 9. In *CBP60G* (IV), motif 9 is absent compared to other branches. It is suggested that motif 8 may be the site where calcium ions bind, based on its position. Motif-related information is in Appendix A. 

### 2.4. Analysis of Cis-Elements in the Promoter of GhCBP60s

The cis-elements in the gene promoter region regulate gene expression by binding to transcription factors and affecting the binding efficiency and stability. Cis-acting elements were predicted in the 2000 bp region upstream of 24 *GhCBP60s* genes. Through the PlantCARE website, a total of four types of cis-acting elements were identified, including light-responsive, hormone-responsive, stress-responsive, and plant development elements (Figure 3). The promoters of all *GhCBP60s* family members contain a large number of light-responsive elements, which are widely distributed in the promoter region, such as the Box4, ACE, G-box, Sp1, AE-box, MRE, CAG-motif, GA-motif, I-box, and GT1-motif. The cis-acting element ABRE is involved in the abscisic acid response, and the cis-acting regulatory elements, the CGTCA-motif and TGACG-motif, involved in methyl jasmonate reactivity are also widely distributed within *GhCBP60s*. Auxin response elements were the TGA-element, AuxRR-core (*GhCBP60A-1*, *AghCBP60C-1A*, *GhCBP60C-2A*, *GhSARD1-1A*, *GhCBP60A-2A*, *GhCBP60A-1*, *GhCBP60B-D*, *GhSARD1-1D*, *GhCBP60E-A*, *GhCBP60E-D*, *GhCBP60C-1D*), and salicylic acid response element TCA-element (*GhCBP60C-2A*, *GhCBP60B-A*, *GhCBP60G-1A*, *GhSARD1-1A*, *GhSARD1-2A*, *GhCBP60A-2A*, *GhCBP60E-D*, *GhSARD1-1D*, *GhSARD1-2D*); in addition, a cis-regulatory NON-box element related to meristem-specific activation was also found in *GhCBP60B_A*. This shows that the *GhCBP60* cotton family plays an essential role in regulating plant response to photoperiod and plant growth and development. 

Among the cis-acting elements related to stress response, ARE elements related to anaerobic induction have the most significant number, TC-rich repeats are related to stress response, MBS are involved in drought induction, and MBS1 cis-acting elements are mainly distributed in the three branches of *CBP60A* (II), *SARD* (III), and *CBP60G* (IV), as well as GhCBP60B. Some cis-elements, such as the LTR and WUN-motif, are responsive to low temperatures and wound response.

### 2.5. Analysis of GhCBP60s Expression in Different Tissues and under Various Stresses

To better understand the role of *GhCBP60s* in cotton growth and development and in abiotic stress, we analyzed expression profile of various tissues of upland cotton (roots, stems, leaves, flowers, sepals, ovules, and fibers) in response to abiotic stress (NaCl, PEG, and low-temperature and high-temperature stresses). The transcriptome data were normalized and visualized (Figure 4). C*BP60A_2A*, *CBP60A_2D*, *CBP60B*, and *CBP60D* are constitutively expressed in various tissues and organs and may play an essential role in plant growth and development. The expression levels of *CBP60D_A* and *CBP60D_D* are relatively high in cotton receptacles, sepals, ovules, and fibers at various stages. The expression of *GhCBP60A_1D* was higher in fibers at 10 and 25 DPA (days post anthesis), and *GhCBP60C_2A* and *GhCBP60C_2D* were higher in fibers at 20 and 25 DPA and in ovules at 20 DPA, indicating that these genes may be related to the termination of fiber development. *SARD* (III) and *CBP60G* (IV) genes are highly expressed in vegetative organs and receptacles, namely sepals and accessory calyces; ovules at 0, 1, and 10 DPA and sepals at 10 and 25 DPA, which may be because of their unique functions (Figure 4A–D).

After stress treatment, the expression level of *GhCBP60* gene changed in most of the tissues. The expression levels of *GhCBP60B_A* and *GhCBP60B_D* increased 12 h after salt stress and PEG stress. The expression levels of *GhCBP60D_A* and *GhCBP60D_D* were increased after stress, while the opposite was seen in *GhCBP60E_D*. The expression levels of *GhCBP60G_1A* and *GhCBP60G_1D* were significantly decreased one hour after low-temperature stress, 12 h after high-temperature stress, and 24 h after salt stress. These findings indicate that these genes might play a role in stress response (Figure 4E–H).

### 2.6. Gene Interaction Network

To analyze the function of the CBP60 protein, we used (STRING v12.0) (https://string-db.org/, accessed on 16 October 2023) to construct a protein–protein interaction (PPI) network (Figure 5). Dystonin, GAS (growth-arrest-specific protein), and related proteins (KOG0516) are at the network’s core by searching for protein families (Figure 5A). Other pathways with higher interaction scores are (KOG3573) caspase, apoptotic cysteine protease, (KOG3544) collagens (type IV and type Xill), and related proteins, such as (NOG008962) non-supervised orthologous group (KOG2687) spindle pole body proteins, which contain UNC-84 domain (KOG1867); Ubiquitin-specific protease (KOG4479); transcription factor e(y)2 (KOG4140); nuclear protein Ataxin-7 (NOG038978); the non-supervised orthologous group (KOG1721), FOG; and zn-finger (NOG007106), a non-managed orthologous group. From these confirmed and predicted interaction pathways, it can be seen that the CBP60 family may be involved in regulating plant growth and development and apoptosis in the nucleus. In the interaction network of the CBP60s family, it was found that no proteins interacted with CBP60D and CBP60E, which may imply that these two proteins perform unique plant functions (Figure 5B). CBP60A interacts with CAMA3 (calmodulin-binding transcription activator 3), PAD4 (lipase-like PAD4), ICS1 (Isochorismate synthase 1), EDS1 (protein EDS1), and FMO1 (probable flavin-containing monooxygenase 1). Experimental evidence indicates that CBP60A interacts with PAD4 and plays a crucial role in plant defense, salicylic acid synthesis, and cell apoptosis. EDS1 (protein EDS1) is the only protein predicted to interact with CBP60B and can trigger early plant defense and hypersensitivity. SARD1 is the core node of the network, with Isochorismate synthase 1 (ISC1), regulatory protein NPR3 (NPR3), transcription factor TGA5 (TGA5), probable WRKY transcription factor 70 (WRKY70), probable flavin-containing monooxygenase 1 (FMO1), CMTA3, senescence-associated carboxylesterase 101 (SAG101), PAD4, and 4-substituted benzoates-glutamate ligase GH3 (GH3.12) implicated in the regulation of basal defense responses to pathogens. Auxin- and salicylic acid-induced transcription are involved in the growth, development, and protection of plants. They have positive effects on functions such as defense against biological damage. 

### 2.7. Functional Analysis and Subcellular Localization of GhCBP60B

Through expression pattern analysis and interaction analysis, we found that the *GhCBP60B* gene is constitutively expressed and might be involved in plant growth and development. We used the VIGS (virus-induced gene silencing) method to conduct functional confirmation in order to better understand the function of *GhCBP60B* in cotton. Ten days after transformation, the newly grown leaves of the positive control plant (TRV::CLA) turned white, showing an albino phenotype (Figure 6A). Forty days after injection, the phenotype of the injected plants was observed and analyzed. The leaves and stems of the TRV::CLA plants all turned white. Compared with the negative control (TRV::00), the plants in the experimental group (TRV::GhCBP60B) showed significant stem shortening and shrunken leaves (Figure 6B,C). We measured the plant height and stem node length of TRV::00 and TRV::GhCBP60B and found that the height of TRV::GhCBP60B plants was significantly lower than TRV::00 (Figure 6D). The aboveground part of the plant was divided into 1–6 nodes from bottom to top according to the petiole. The length of nodes 4–6 of the TRV::GhCBP60B plant was significantly shorter than that of the TRV::00 plant, which also caused the TRV::GhCBP60B plant to be shorter than the TRV::00 plant (Figure 6E). As shown in Figure 6F, in the qRT-PCR analysis of TRV::00 and TRV::GhCBP60B, the expression levels of the silenced plants were significantly reduced compared with the negative control.

In order to study whether this phenomenon of TRV::GhCBP60B plants is related to autoimmunity, we performed qRT-PCR on some essential immune marker genes. After GhCBP60B was silenced, the expression levels of genes such as PR1, PR2, and SID2 were significantly reduced (Figure 6G–I). Finally, we also conducted the subcellular localization of the GhCBP60B gene in tobacco leaves and found that it was consistent with the prediction results of the website, and GhCBP60B_A and GhCBP60B_D are only expressed in the nucleus (Figure 7).

### 2.8. The Silencing of the GhCBP60B Gene Reduces Cotton Resistance to Drought and Salt Stress

In order to explore the potential functions of the *GhCBP60B* gene under various abiotic stress conditions in depth, we used virus-induced gene silencing VIGS technology to silence the *GhCBP60B* gene in cotton specifically. About ten days after transformation, it was observed that the positive plants (TRV::CLA) exhibited significant albino traits (Figure 8A). This phenotype indicated that the *GhCBP60B* gene had been successfully silenced. Subsequently, about two weeks after inoculation, we applied two stress treatments of NaCl and PEG to the control plants (TRV::00) and the experimental group plants (TRV::GhCBP60B). As shown in Figure 8B,C, under drought and salt stress conditions, the leaf shrinkage of silent plants was more evident than that of control plants. To confirm the efficiency of *GhCBP60B* gene silencing, a qRT-PCR analysis was performed. The results showed a decreased expression level of the *GhCBP60B* gene in silenced plants compared to control plants, indicating the effective silencing of the *GhCBP60B* gene in the silenced plants (Figure 8D). To further explore the mechanism of the *GhCBP60B* gene, in response to drought and salt stress, we measured the POD (peroxidase) activity of plants treated with 20% PEG 6000 and 200 mM NaCl for 24 h. The results showed that after stress treatment, the POD activity of silenced plants was significantly lower than control plants (Figure 8E,F). This finding suggests that silencing the *GhCBP60B* gene may weaken the physiological tolerance of cotton plants to drought and salt stress, potentially impacting their ability to survive under stressful conditions.

## 3. Discussion

Calcium ions (Ca^2+^) play a crucial role in plants. It is ubiquitous in plant cells and serves as a critical second messenger involved in plant defense response signaling pathways. In addition, calcium ions play an indispensable role in plants’ selective absorption of ions, signal transmission, and improvement of stress resistance [13,14,15]. The *CBP60* family, a unique family of plant proteins, plays a key role in various stresses tolerance of plants. Most *CBP60* family members are directly involved in plant disease resistance, immune mechanism, and salicylic acid (SA) accumulation. These functions together enhance the plant’s ability to adapt to the environment [16,17]. Under normal growth conditions, *CBP60b*, as a key member of *CBP60*, is responsible for maintaining the essential immune function of plants. It is worth mentioning that *CBP60b* shows high expression levels in various tissues and growth stages of plants. In contrast, other members of this subfamily are expressed at lower levels and are usually restricted to the root tip. These observations emphasize the dominant role of *CBP60b* as a *CBP60* subfamily during plant growth [6].

Although the functions of the CBP60 family in plants have been partly elucidated, particularly in response to disease resistance, immunity, and salicylic acid accumulation, their precise roles in plant growth, development, and response to abiotic stressors remain enigmatic. By identifying *CBP60* family members from various plants, this study systematically analyzed the developmental relationships, chromosomal location, gene architecture, regulatory elements, conserved regions and motifs, and the complex structure of the protein of *GhCBP60*. In addition, we further explored the expression pattern of *GhCBP60*. This study provides strong evidence for the vital role of *GhCBP60B* in cotton growth and development and abiotic stress.

In the identification and phylogenetic analysis of *CBP60* gene family of more than 20 plants, the *CBP60* family was divided into four branches. These 20 plants include algae, mosses, ferns, monocotyledons, and dicots. (Figure 1) No *CBP60* family members have been found in algae. *CBP60* family members first appeared in ferns and mosses, and all appeared in the first branch, indicating that *CBP60B-F* members are the original members of the *CBP60* family. *CBP60A*, *CBP60G*, and *SARD1* are all differentiated from *CBP60B-F*. According to the member distribution of each branch of dicotyledons and monocotyledons, it was noticeable that evolution speed of subfamilies II, III, and IV is much faster than that of the prototype group (I). As the earliest angiosperm, *CBP60* family members were identified in Atr1, belonging to types I, II, and IV, respectively, but lacking *SARD*(III), indicating that the SARDI subfamily was differentiated after the basal angiosperms, these findings are consistent with previous research [18]. Of the four types of cotton, 12 members of the *CBP60* family were identified in diploid Asian cotton and *G. raimondii* and 24 *CBP60* family members were placed in each of the tetraploid cotton species *G. hirsutum* and *G. barbadense*, indicating that the *CBP60* family members are relatively comparable in cotton and no loss occurs during gene duplication. Chromosome analysis showed that there are multiple *CBP60* family members on the A08, D08, A12, D12, A13, and D13 chromosomes of upland cotton, while there is only one or none on other chromosomes, indicating that the *CBP60* family genes are unevenly distributed in cotton (Appendix A). In upland cotton, the number of exons of the *CBP60* gene is also different. It has the same number of exons in the same branch. Branches I and IV have eight exons, while branches II and III only have seven exons. The number of motifs within the same branch of *GHCBP60s* members are highly conserved, which might be the functional similarity that they perform (Figure 2).

To illustrate the critical role that *GHCBP60* gene plays in response to biotic and abiotic stress, the results of cis-acting element analysis showed that hormone response-related elements: the abscisic acid-responsive cis-element ABRE and the methyl jasmonate-responsive cis-acting element. The regulatory elements, the CGTCA-motif and the TGACG-motif; the auxin response element, TGA-element; the AuxRR-core; and the salicylic acid response element, TCA-element, are widely distributed in *GhCBP60s* (Figure 3). In addition, predicted cis-elements are also related to the meristem in *GhCBP60B_A* and are involved in the specific activation of the relevant cis-regulatory element NON-box. This indicates that *CBP60* family members may also be related to plant growth and development. These results are consistent with studies on soybean and eggplant [9,19]. Transcriptome analysis results show that the *GHCBP60B* gene is constitutively expressed in cotton, and the *GHCBP60C* gene is predominantly expressed in specific tissues. *GHCBP60C_2A* and *GHCBP60C_2D* are highly expressed in 20 and 25 DPA fibers and 20 DPA ovules. We speculate that *GHCBP60* gene may be involved in fiber termination development (Figure 4A). After stress, the expression levels of most *GhCBP60s* genes changed. Among them, the expression levels of *GhCBP60A_2A* and *GhCBP60A_2D* increased after stress, indicating that *CBP60A* also plays an essential role in abiotic stress resistance (Figure 4B), which is consistent with the findings of previous research [9,19]. The expression levels of *GhCBP60B_A* and *GhCBP60B_D* showed an upward trend after salt stress and PEG stress, implying that *GhCBP60B* plays a role in cotton in response to salt stress and PFG stress, but few studies are concerned with this.

We further studied the role of *GhCBP60B* in cotton growth and development and response to abiotic stress (Figure 6). Experimental results show that after silencing the *GhCBP60B* gene in cotton for 50 days, cotton will have obvious plant dwarfism, shrunken leaves, and shortened stem nodes. The previous cis-acting element analysis also found that *GhCBP60B* has a meristem-specific, activation-related, cis-regulatory NON-box element. Expression pattern analysis also suggested that *GhCBP60B* continuously expressed in cotton. Li et al. (2021) found that overexpression and interference with the *CBP60B* gene would cause shorter stems and smaller leaves in *Arabidopsis*, which is consistent with our conclusion [6]. We also found that after the silencing of *GhCBP60B*, the expression levels of immune marker genes PR1, PR2, and SID2 were significantly reduced. The *Arabidopsis CBP60B* gene silencing also caused changes in the expression levels of immune genes, indicating that the phenotypes of TRV::GHCBP60B cotton may be related to immune activities in the plant (Figure 8). After silencing the *GhCBP60B* gene by using the TRV-VIGS method, we observed that the tolerance of the affected cotton plants to salt and drought environments was significantly reduced, which was obvious evidence through the emergence of sensitive phenotypes and abnormal leaf development. Further physiological indicators were tested, and it was found that compared with TRV::00 plants, the activities of SOD and POD enzymes in the leaves of the silenced plants were significantly reduced. Plants have formed an efficient enzyme antioxidant defense system, and enzymes such as SOD and POD play an important protective role. SOD and POD enzymes can avoid or minimize damage caused by abiotic stress. The decreased activity of SOD and POD enzymes indicates that the *GhCBP60B* gene plays a positive role in plant response to adverse stress and leaf development. In previous studies, *CBP60B* was significantly involved in the immune functions of plants and has been shown to be a major *CBP60* that maintains primary immunity and responds to the invasion of pathogenic bacteria [6,20]. Our results show that *GhCBP60B* plays a vital role in cotton’s growth and development process and in resisting abiotic stress.

## 4. Materials and Methods

### 4.1. Identification of CBP60 Protein Family Members

The *Arabidopsis CBP60* family protein sequence was obtained from the Arabidopsis Information Resource Database and from phytozome13 (https://phytozome-next.jgi.doe.gov/, accessed on 3 July 2023). *Amborella trichopoda* (version 1.0), *Ananas comosus* (version 3.0,) *Glycine max* (version 1.0), *Marchantia polymorpha* (version 3.0), *Oryza Sativa L* (version 7.0), *Populus trichocarpa* (version 4.1), *Physcomitrium patens* (version 3.3), *Santa Cruz* (version 2.1), *Solanum lycopersicum* (version5.0), *Sorghum bicolor* (version 5.1), *Theobroma cacao* (version 2.1), *Vitis vinifera* (version 2.1), Vunguiculata (version 1.0), Cpversion 2.1), *Volvox criteria* (version 2.1), and the genome sequence of *Zea mays* (version 4.0) were used to obtain sequences from phytozome13. The genome sequence data of four cotton species, namely *Gossypium arboreum* (CRI, version 1.0) [21], *G. raimondii* (JGI, version 2.0) [22], *G. hirsutum* (ZJU, version 2.1) and *G. barbadense* (ZJU, version 1.1) [23], were extracted from the COTTONGEN website (http://www.cottongen, accessed on 5 July 2023) [24] and the Cotton Functional Genomics Database CottonFGD (accessed on 5 July 2023) [25] and downloaded.

We performed BLASTp on the four cotton protein sequences using the amino acid sequence of Arabidopsis *AtCBP60s* as a query sequence and preliminarily determined the candidate sequences of other selected species. Subsequently, we used the Pfam website (http://pfam.xfam.org/, accessed on 3 July 2023) to search for Pfam ID (PF07887) and downloaded the relevant documents [26], and then used HMMER v3 and the hidden Markov model (HMM) to search the homologous sequences, which were further verified through the SMART website (http://smart.embl.de/, accessed on 12 August 2023) and, finally, intersected with the previous candidate sequences to determine the members of the *CBP60* family in cotton [27]. We used the same method to identify CBP60 family members in *Amborella trichopoda*, *Ananas comosus*, *Cpapaya*, *Glycine max*, *Marchantia polymorpha*, *Oryza Sativa* L., *Populus trichocarpa*, *Physcomitrium patens*, *Santa Cruz*, *Solanum lycopersicum*, *Sorghum bicolor*, *Theobroma cacao*, *Vitis vinifera*, *Vunguiculata*, *Volvox carter*, and *Zea mays.* All sequences and registration numbers are shown in Appendix A.

In addition, the ExPASy tool (http://web.expasy.org/, accessed on 28 August 2023) was used to analyze the physicochemical properties (i.e., length, molecular weight, and isoelectric point) of CBP60 family proteins in upland cotton through the Cell-PLOC online website (http://www.csbio.sjtu.edu.cn/bioinf/Cell-PLoc-2/l, accessed on 12 August 2023) [28]. We used the ClustalW program to accurately align CBP60 protein sequences to ensure accurate alignment between sequences [29]. Subsequently, with the help of MEGA11.0 software, we constructed a phylogenetic tree based on the neighbor-joining algorithm. We conducted a bootstrap test with 1000 replicates to enhance the robustness of the analysis [30].

### 4.2. CBP60 Chromosomal Localization

Visualizing the position information of the *CBP60* gene on chromosomes was completed from the MG2C (Map Gene 2 Chromosome, v2.1, http://mg2c.iask.in/mg2c_v2.1/, accessed on 12 September 2023) website [31,32].

### 4.3. Analysis of the Conserved Protein Motifs and Gene Structure

The online software MEME (http://meme-suite.org/tools/meme, accessed on 15 September 2023) was used to analyze the conserved motif of the sequence [33], and the online gene prediction tool GSDS v2.0 (http://gsds.cbi.pku, accessed on 20 September 2023) was used for gene structure analysis. The phylogenetic tree maps [34], exon/intron structure, motifs, and conserved domains of CPBP60 in cotton and *Arabidopsis thaliana* were completed by Tbtools [35].

### 4.4. Analysis of Cis-Elements in the Promoter Regions of GhCBP60s

The base sequence of the 2000 bp upstream of the *GhCBP60s* gene was obtained as promotors from the online website CottonFGD [25]. Cis-element analysis was performed using the PlantCare website (http://bioinformatics.psb.ugent, accessed on 25 September 2023) and visualized using TBtools [36].

### 4.5. Expression Analysis of GhCBP60s

The RNA-seq data of different tissues at different growth stages and in response to abiotic stresses (salt, drought, cold, and heat) of upland cotton were obtained from the cottonfgd website. The heat map of the expression pattern of *GhCBP60s* was constructed using the fragments per kilobase of exon model per million mapped fragments (FPKM) value through the online website ImageGP. (https://www.bic.ac.cn/ImageGP/index.php/Home/Index/PHeatmap.html, accessed on 18 September 2023).

### 4.6. RT-qPCR Validation

FastPure^®^ Universal Plant Total RNA Isolation Kit (Vazyme Biotech Co., Ltd., Nanjing, China) was employed to extract RNA following the instructions. Subsequently, RNA purity, degradation, and concentration were determined by NanoDrop 2000 spectrophotometer (NanoDrop, Wilmington, USA) and 1% agarose gel electrophoresis. Total RNA was reverse-transcribed to cDNA with PrimeScript™ RT Master Mix (Tli RNaseH Plus) (Takara Biomedical Technology Co., Ltd., Beijing, China). The specific primers for these genes were designed using the online website cottonGVD. In quantitative real-time polymerase chain reaction (RT-qPCR), the gene *GhActin* was employed as an internal reference, and a total of three biological replicates and three technical replicates were established for each gene. Genes expression levels were normalized in comparison with the expression level of the *Actin* gene and quantified using the 2^−ΔΔCt^ method based on Ct values [37]. The primers utilized for the analysis can be found in the Appendix A. Hieff^®^ qPCR SYBR Green Master Mix was used to perform qRT-PCR on the bio-rad CFX96 fluorescence quantitative PCR platform, and raw data were collected using Bio-Rad CFX Manger software (version 2.1). Primers, reaction systems and RT-qPCR steps are described in Appendix A.

### 4.7. Virus-Induced Gene Silencing of GhCBP60B in Upland Cotton

Zhong100 was used as an experimental material and planted in the controlled chamber. After de-linting and washing the seeds with concentrated sulfuric acid, healthy and plump seeds were selected and cultured for 16 h of light (12,000 Lx, 28 ℃) and 8 h of darkness (0 Lx, 25 ℃). A knock-down vector VIGS system based on the tobacco rattlesnake virus (TRV) was constructed to silence *GhCBP60B* [38]. The silencing fragment of *GhCBP60* was designed through the SGN VIGS tool (https://vigs.solgenomics.net/tdsourcetag=spcqq_aiomsg, accessed on 30 September 2023) and cloned into the pTRV2 vector using specific primers. Single colonies of freshly cultured TRV1, TRV2 containing inserted fragments, empty TRV2- and TRV2-inserted CLA *agrobacterium* fragments, were taken and inoculated into 3 mL LB (kanamycin, 100 μg/mL; rifampicin, 50 μg/mL) and kept cultured at 28 °C and 170 rpm for 16 h. Then, the 1% of the infusion volume was added to 50 mL of LB (Kanamycin, 100 μg/mL; rifampicin, 50 μg/mL), at 28 °C and 170 rpm, and was cultured for 12 h, until the bacterial solution OD600 reached to 0.6. Then, the liquid culture was centrifuged at 4000 rpm for 10 min to collect bacterial cells and resuspended with an appropriate volume of resuspension solution (10 mM MgCl2, 10 mM MES, and 200 uM acetosyringone) to a final concentration of 2.0 (OD600), and the suspension was placed in the dark at 28 °C for 3 h. Then, the TRV1 solution and the solution containing negative control (TRV::00), positive control (TRV::CLA) (*Cloroplastos alterados*), and silenced gene (TRV::*GhCBP60B*) were suspended at a ratio of 1:1 (*v*/*v*). After mixing, the seven-day-old cotton cotyledons were inoculated. Plants obtained after gene silencing were cultured at 23 °C with a 16 h/8 h light/dark cycle. Two weeks after the inoculation with bacterial solution, the inoculated cotton plants were treated with 20% PEG6000 and 200 mM NaCl solution for two days. The ultrapure water was used as a control. Three biological replicates and three technical replicates were set up, with at least ten strains in each experimental group. The GhCBP60B gene-specific primer sequences of VIGS are shown in Appendix A.

### 4.8. Subcellular Localization of GhCBP60B

The pCAMBIA1300-GFP vector was used to fuse the full-length coding sequences of GhCBP60B_A and GhCBP60B_D with the stop codon removed, and then the constructs were transformed into Agrobacterium GV3101. The infiltration buffer containing the fusion constructs 35S::GhCBP60B_A:GFP and 35S::GhCBP60B_D:GFP (final concentration of MES 10 mM, MgCl_2_ (20 mM), acetosyringone (10 µM) was injected into the tobacco leaves through a needle-free syringe. After three days of incubation, the epidermis was removed and the subcellular localization of GhCBP60B_A and GhCBP60B_D proteins was observed using fluorescence microscopy. Transformed tobacco cells were visualized by using the 20× objective of the AXIO IMAGER A2 fluorescence microscope, combined with bright field, GFP (excitation/emission: 488/498–548 nm), and DAPI (excitation/emission: 405/421–523 nm) filters.

### 4.9. Determination of POD and SOD Contents

Leaf samples were obtained from silenced plants and negative controls before and after PEG and NaCl treatments. These samples were promptly frozen in liquid nitrogen to maintain their biochemical activity. Subsequently, we used special detection kits (Solaibao Science & Technology Co., Ltd., Beijing, China) for POD (BC0095) and SOD (BC0175) to accurately determine the enzymatic activities of POD and SOD in these samples, following the previously described detection methods as well as the instructions provided in the kit manuals [39].

## 5. Conclusions

In this study, we identified a total of 162 *CBP60* genes in 21 plant species, including 12, 12, 24, and 24 genes in the genomes of *Gossypium arboretum*, *G. raimondii*, *G. hirsutum*, and *G. barbadense,* respectively. We systematically explored the various characteristics of the *CBP60* gene in upland cotton, including its phylogenetic relationship, gene structure, conserved motifs, cis-elements, and protein–protein interaction network, as well as analyzing its spatiotemporally specific expression pattern. Further experimental studies showed that the silencing of the *GhCBP60B* gene significantly affects the growth and development of cotton, leading to phenotypic changes such as plant dwarfing, leaf shrinkage, and stem node shortening. In addition, our study also found that the *GhCBP60B* gene can enhance cotton’s resistance to drought and salt stress by regulating peroxidase activity. These findings provide new perspectives and strategies for using the *GhCBP60B* gene to improve cotton plant types and improve its drought and salt tolerance.

## Figures and Tables

**Figure 1 ijms-25-04349-f001:**
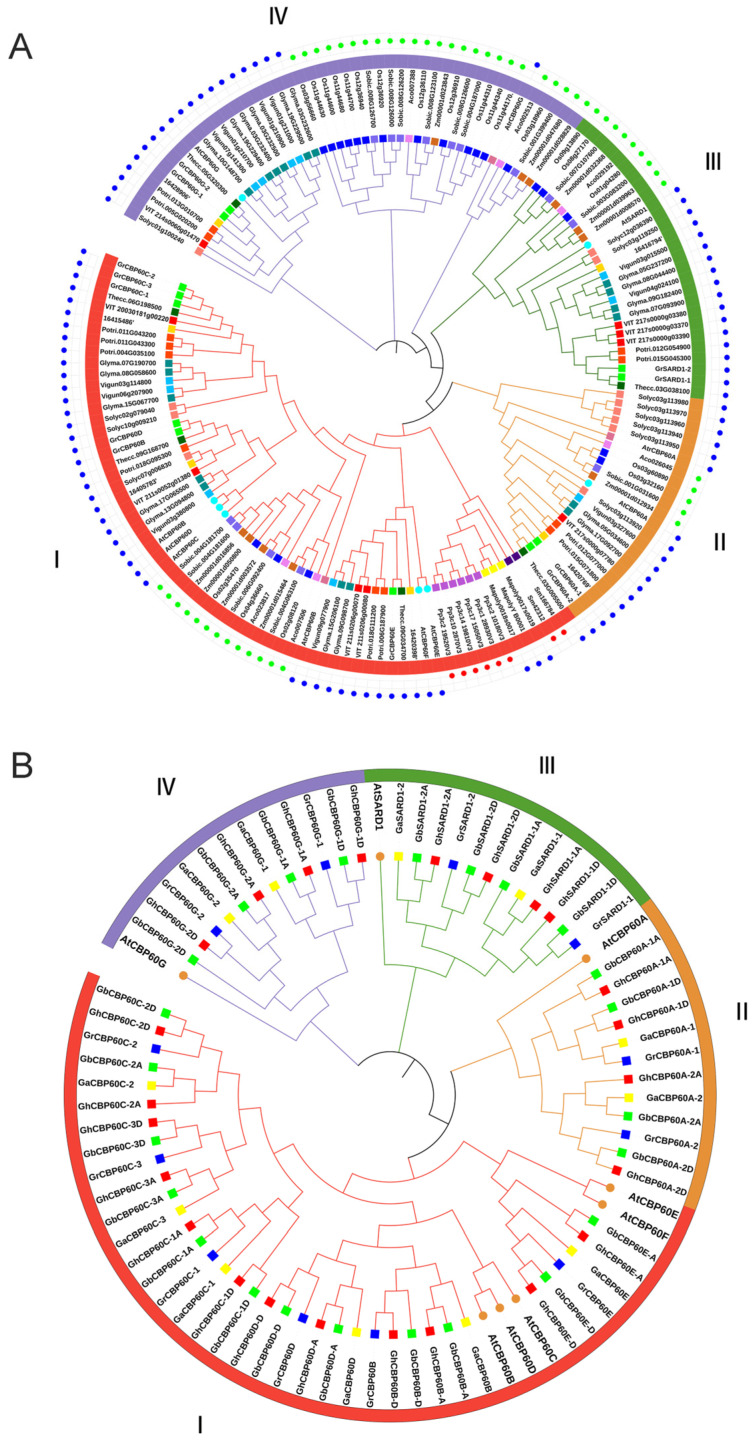
The phylogenic trees of *CBP60* family genes. (**A**) Phylogenetic relationship of the 162 identified *CBP60* genes from 21 plant species. (**B**) Phylogenetic relationship of the 72 identified *CBP60* genes from four cotton species and Arabidopsis. The two neighbor-joining (NJ) phylogeny trees were constructed using MEGA 8.0 software. The blue dots in the outer ring of (**A**) represent the dicotyledonous clock, the green dots represent the monocot clock, and the red dots represent the fern and moss clocks.

**Figure 2 ijms-25-04349-f002:**
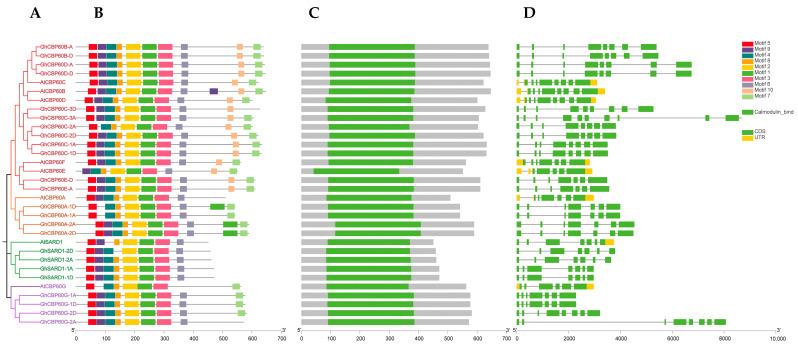
Comparison of the gene structure, conserved protein motifs and domains in *CBP60* genes between *Arabidopsis* and *Gossypium hirsutum* (**A**) The NJ phylogenetic tree was constructed based on the full-length sequences of *Arabidopsis* and *Gossypium hirsutum CBP60* proteins using MEGA 7.0 software. (**B**) The motif composition of Arabidopsis and *Gossypium hirsutum CBP60* proteins. The motifs, numbered 1–10, are displayed in different colored boxes. (**C**) Schematic representation of the conserved domains in *Arabidopsis* and *Gossypium hirsutum CBP60* proteins. (**D**) Exon–intron structure of *Arabidopsis* and *Gossypium hirsutum CBP60* genes.

**Figure 3 ijms-25-04349-f003:**
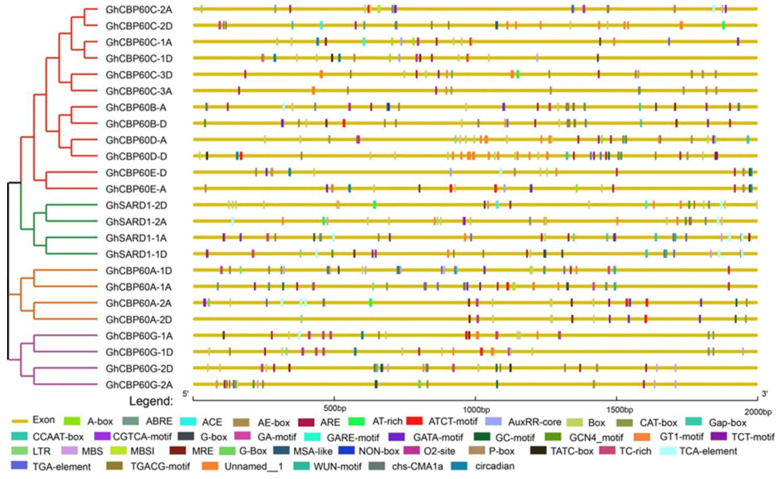
Analysis of cis-acting elements in the upstream promoter region of *GhCBP60s*. Differently colored boxes represent unique identified cis-elements.

**Figure 4 ijms-25-04349-f004:**
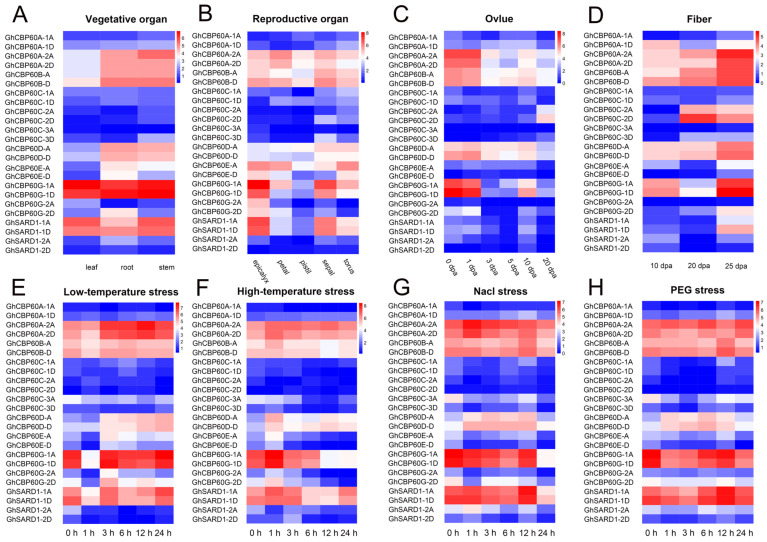
Expression patterns of *GhCBP60* genes in multiple tissues (vegetative organs (**A**), reproductive organs (**B**), seeds/ovules (**C**), and fibers (**D**)) and different abiotic stress treatments (salt (**E**), drought (**F**), low temperatures (**G**) and high temperatures (**H**)) of upland cotton. The red in the legend represents high expression levels, and the blue indicates low expression levels. The expression heat map was generated based on the logarithms of the FPKM values.

**Figure 5 ijms-25-04349-f005:**
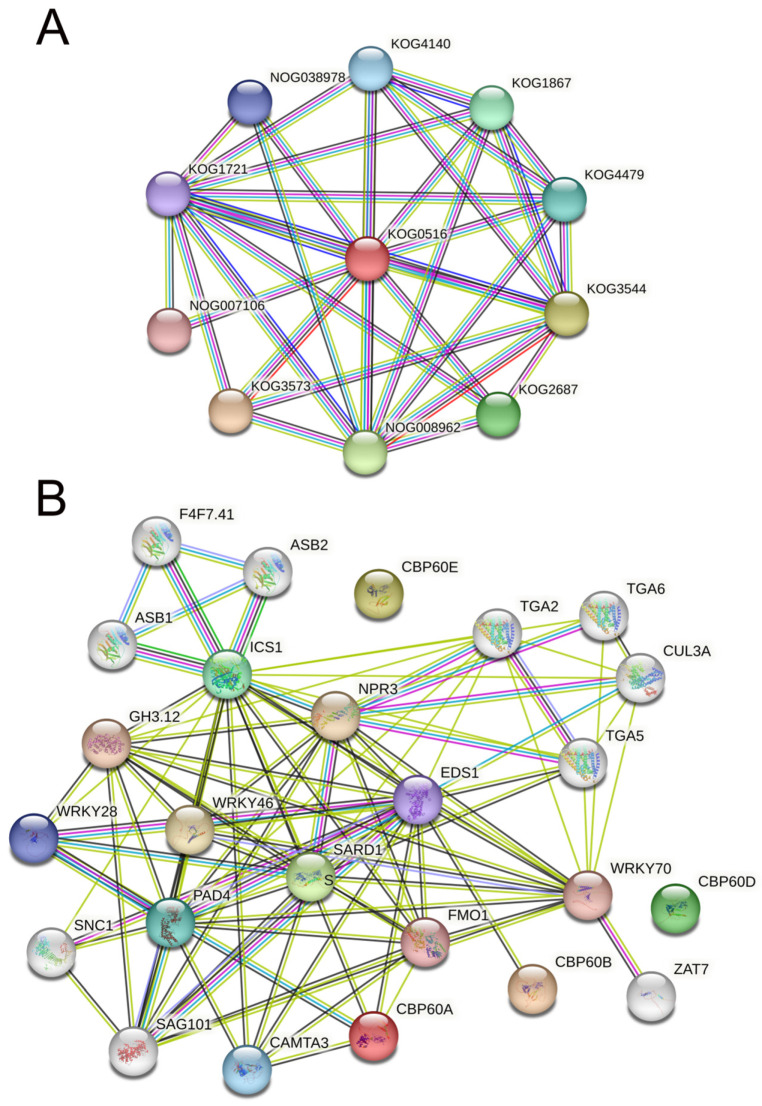
Interaction network of CBP60 proteins. (**A**) Interaction network of CBP60 protein families. (**B**) Interaction network of GrCBP60 proteins with other proteins. The light green lines represent protein–protein interactions based on text mining; dark green lines represent protein–protein interactions based on gene neighborhood; black lines represent protein–protein interactions based on co-expression; blue lines represent protein–protein interactions based on gene co-occurrence; and purple line represents protein–protein interactions based on protein homology.

**Figure 6 ijms-25-04349-f006:**
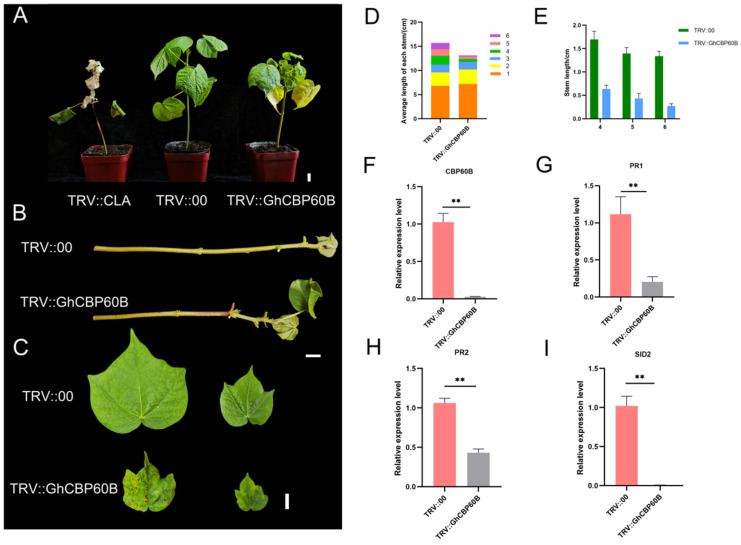
Effects of *GhCBP60B* gene silencing on cotton plant growth and development. (**A**) Phenotypes of positive control (TRV::CLA), negative control (TRV::00), and *GhCBP60B* gene-silenced transgenic plants (TRV:: GhCBP60B) under normal conditions. (**B**) Main stem of TRV::00 and TRV::GhCBP60B plants. (**C**) TRV::00 and TRV::GhCBP60B plants that had just fully expanded their young and mature leaves. (**D**) The length distribution of each stem node in TRV::00 and TRV::GhCBP60B plants. (**E**) Statistics on the length of stem nodes 4, 5, and 6 in TRV::00 and TRV::GhCBP60B plants. (**F**–**I**) Relative expression levels of GhCBP60B, PR1, PR2, and SID2 in TRV::00 and TRV::GhCBP60B plants. The GhActin gene was used as the internal reference. ** indicates *p* < 0.01. In (**E**–**I**) Three biological replicates were performed and data are the means ± standard deviation.

**Figure 7 ijms-25-04349-f007:**
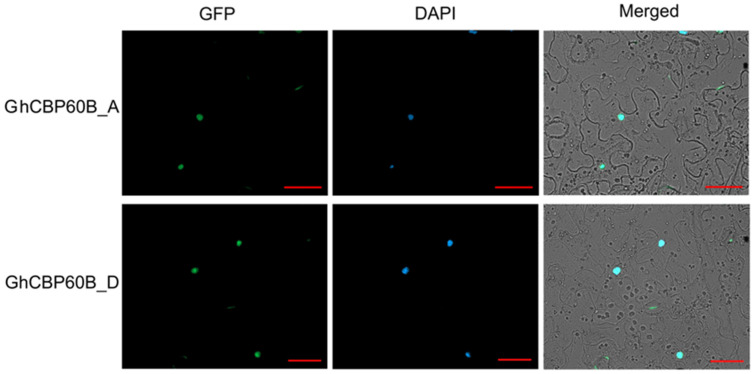
Subcellular localization of GhCBP60bB_A and GhCBP60bB_D. DAPI: a DNA dye that localizes to the nucleus and emits blue fluorescence. Bar = 20 μM.

**Figure 8 ijms-25-04349-f008:**
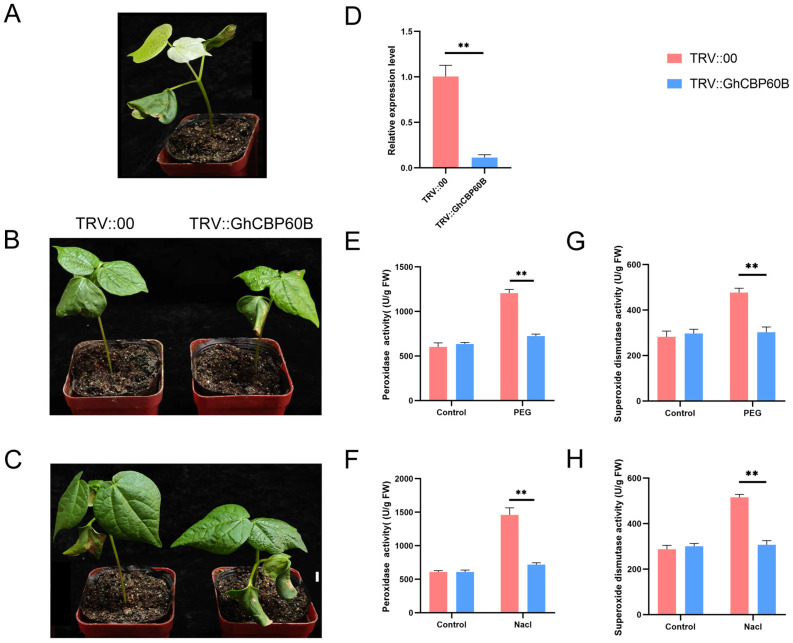
Silencing of the *GhCBP60B* gene reduced the tolerance of cotton plants to drought stress and salt stress. (**A**) Albino phenotype plant (TRV::GhCLA; positive control). Phenotypes of negative control (TRV::00) and *GhCBP60B* gene-silenced transgenic plants (TRV::GhCBP60B) under drought (**B**) and salt stress (**C**) conditions for 48 h. (**D**) Relative expression levels of *GhCBP60B* in TRV::00 and TRV::GhCBP60B plants. The GhActin gene was used as an internal reference. TRV::00 and TRV::GhCBP60B plants peroxidase (POD) activity after 48 h PEG (**E**) and salt stress (**F**) treatment. TRV::00 and TRV::GhCBP60B plants superoxide dismutase (SOD) activity after 48 h PEG (**G**) and salt stress (**H**) treatment. Error bars represent standard deviations estimated from three independent experiments. ** indicates *p* < 0.01. In (**D**–**H**) Three biological replicates were performed and data are the means ± standard deviation.

## Data Availability

The original contributions presented in the study are included in the article/Appendix A, further inquiries can be directed to the corresponding author/s.

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
