# Peer review of "Genome-Wide Identification of Calmodulin-Binding Protein 60 Gene Family and the Function of GhCBP60B in Cotton Growth and Development and Abiotic Stress Response"

_ijms, 2024, doi:10.3390/ijms25084349_

Round 1

Reviewer 1 Report

Comments and Suggestions for Authors

The work discusses important aspects of identifying gene families responsible for abiotic stresses. The authors clearly introduce the reader to the issue, presenting a goal that can be interpreted in a broader context. The literature additionally supports the theses put forward by the authors and justifies the research goal. The authors describe in detail the methodology of laboratory tests and secondary analyzes of data from database websites, which confirms their broad approach to the topic. The work, supported by previous results of other researchers, constitutes an important contribution to the development of a given field of science. The discussion and conclusions constitute an adequate summary of the work, although it is possible to consider extending the conclusions to include the consistency of the obtained results with other studies.

please make changes:

In Figures 8E and 8F, please keep the Y axis scale the same, which will allow the reader to better compare the values in the graphs.

In the literature, in position [9], on line 570, please change the year from 2003 to 2023.

Reviewer 2 Report

Comments and Suggestions for Authors

Authors performed generally import study but the quality of presented manuscript is low and it should be strongly improved (details below). There is many grammar and typographical errors. Also more markers of a drought and salt stress response should be used and presented (not only POD), now these results are incomplete -presented results could not suport conclusions. Introduction, Materials and methods and Discussion section should be improved. Precise description how to improve the manuscript is below:

1. Lines 13-17, 18, 86, 91, 92, 102-107; italicize plant, animal or bacteria Latin names. Check the entire text.

2. Introduction section – write more of PR1, PR2 and other immunity related genes in context of GhCBP60B function.

3. 47,48- Ca ion write 2+ in upper index (Ca2+), check the whole text

4. Lines 11-121 rewrite the sentence

5. Line 112- what is Sm?

6. Line 145- should be correspond

7. Section 2.3 provide the sequences of conserved motifs 1-10, as a suplement file. Write in the suplement file what putative functions could they play?

8. Rewrite the section 4.7 ; do not start the sentence with a number, for example 2 but a word two, do not write „Set up three biological replicates and three technical replicates, with at least ten strains in each experimental group’, instead write for example „Three biological replicates and three technical replicates were set up, with at least ten strains in each experimental group”.  

Provide information/meaning of the „CLA”? What function play CLA in Agrobacterium?

Do not write ug but µg.

Write that VIGS is a gene silencing method

9.Line 288-290, instead of fluorescence quantitative analysis write qRT-PCR, it will be more clear to readers.

10. Rewrite sentence „In the fluorescence quantitative analysis of 288 TRV::00 and TRV::GhCBP60B, as shown in Figure 6F, the expression level of the silenced plants was significantly reduced compared with the negative control”.

11.Lines 303-306; there is no precise description of immunofluorescence experiment, equipment (model name, manufacturer, country of origin), staining details, used magnification in material and methods section, provide it

12. Section 2.8 Why the POD was choosen as a drought and salt stress response marker? Why not other markers of these stresses?

13. Discussion section

There is no discussion of the role of essential immune marker genes such as PR1, PR2, and SID2

Line 421-424 Authors write of SOD, but results concerning SOD were not presented in results section.

14. Section 4.6; provide the information concerning:

1.       How the quality of RNA was assured?

2.       Amount of RNA per one qPCR sample

3.       Details of RT and PCR step; temperature, duration, composition of reaction mixture

4.       Name and manufacturer of equipment used to qPCR

5.       Software used to acquire raw qPCR data

6.       Expected length of PCR products

7.       Actin gene from which plant?

15. Typhos; remove dot before square bracket in line 342 and after S and A in SA in line 345. Line 512- what is „kana”? Use Kan or kanamycin. Line 515- write MgCl2 not MgCl2. Check the whole text.

Comments on the Quality of English Language

Extensive editing of English language required.

Reviewer 3 Report

Comments and Suggestions for Authors

The article headlined “Genome-wide identification of calmodulin-binding protein 60 gene family and the function of GhCBP60B in cotton growth and development and abiotic stress response” is dedicated to the important topic – the search of the effective candidate genes to increase a plants tolerance to the drought and salt stress. Authors have analyzed a big source of data and were able to identify evidence for the role of GhCBP60B in cotton growth, development and resistance to abiotic stress. During the study authors identified CBP60 family members from various plants, as well as they analyzed the developmental relationships, chromosomal location, gene architecture, regulatory elements, conserved regions and motifs, and the complex structure of Ghcbp60 protein. Authors have obtained interesting results which could be used in future genetic engineering or gene editing projects aimed to the crops improvement.

There are several critical points, which should be explained in details.

From the data presented in the manuscript it is hard to say, that authors obtained dwarf plants. Yes, leaves smaller and stem shorter in comparison with control plants, but I suggest not to use a dwarf term. Or authors should give more data proving dwarfism but not just decrease in organs size.

The qPCR should be described more detailed. What KIT was used for RNA isolation? How RNA was reverse transcribed (using oligo dT or random hexamer primers)? What was the efficiency of qPCR primers and how it was taken into account during ΔΔCt calculation? What KIT was used for qPCR? How did you control the absence of gDNA in RNA fraction? Did you make DNAse treatment of RNA fraction? Did you set up a control PCR with RNA as a template to check gDNA absence?

The text is written in very brief and fast style, has a lot of punctuation mistakes and needs to be read carefully to correct all the mistakes. Authors use different tense in different parts of the article. Please unify the tense, for example to the past tense.

In articles dealing with phylogenetics usually posterior probability or bootstrap values should be presented at the phylogenetic trees. Please provide this information.

Methods of SOD and POD activities measurements are missing in the text, as well as a results regarding SOD. Please add.

Please check all the genes names. They should be written using Italic type.

Line 36: “Calciumions” change to “Calcium ions”.

Line 45: “Calmodulin, CaM)” change to “Calmoduli (CaM)”

Line 47: “Ca2+” change to “Ca2+”. The same applies for the entire text.

Line 53: “Arabidopsis” change to “Arabidopsis” (Italic type). The same applies for the entire text.

Line 80: “Plays” change to “plays”

Line 86: “Arabidopsis thaliana” change to “Arabidopsis thaliana” (Italic type).

Lines 91-92: Scientific Latin names of the species should be written using Italic type.

Figure 1: Hard to read. Please increase the size of the trees. Probably it is better to orientate the A and B figures vertically (A on the top and B at the bottom).

Lines 102-107: Scientific Latin names of the species should be written using Italic type. Moreover, it is better to give full scientific names of the species, including date and descriptor name.

Figure 2: The text at the picture is unreadable. Please increase type size or figure scale.

Lines 173-180: Please check the Italic type for all Latin species names.

Please describe what is DPA. There is only abbreviation in the text. “Days post anthesis (DPA)”. The same applies for “VIGS”.

Figure 4: Also hard to read text at the figure. Is there a chance to increase the type a little bit?

Figure 8: There are no “A”and “D” description under the figure.

Line 342: Please remove the period in “stress resistance.”.

Line 430: Change “bacteria. (6,20)” to “bacteria (6,20).”.

Please add the accession numbers to all the sequenced used in the study.

Line 51: What is “Kana”? Kanamycin? Please use full name of antibiotic like you did for rifampicin.

Line 512: Change “L.B.” to “LB”.

Please check the part “4.7. Virus-induced gene silencing of GhCKX14 in upland cotton” for the periods in the end of the sentences.

Line 509: Check the sentence to the consistency (especially for the word Agrobacterium).

Comments on the Quality of English Language

Authors use some scientific slang which could be corrected. The tense used in the text should be unified. 

Round 2

Reviewer 2 Report

Comments and Suggestions for Authors

Authors corrected the manuscript according to suggestions. Manuscript is meaningfully improved and I have no significant comments. Only small typhos or grammar errors could be improved:

1. Write names of genes in italics ; line 483 and 508.

2. Rewrite the sections 4.7, for example as below (in red added fragments), it could be easier to read the text.:

Section 4.7

Zhong100 was used as an experimental material and planted in the controlled chamber. After de-linting and washing the seeds with concentrated sulfuric acid, healthy and plump seeds were selected and  cultured for 16 hours of light (12000 Lx, 28 ) and 8 hours of darkness (0 Lx, 25 ). A knock-down vector VIGS system based on the tobacco rattlesnake virus (TRV) was constructed to silence GhCBP60B [39]. The silencing fragment of GhCBP60 was designed through the SGN VIGS tool  (https://vigs.solgenomics.net/ tdsourcetag=spcqq_aiomsg) and cloned into the pTRV2 vector using specific primers. Single colonies of freshly cultured TRV1, TRV2 containing inserted fragments, empty TRV2 and TRV2 inserted CLA fragment agrobacterium, were taken and inoculated into 3 ml LB (kanamycin, 100 μg/ml; rifampicin, 50 μg/ml) and kept cultured at 28 and 170rpm for 16 hours. Then the 1 % of the infusion volume was added to 50 ml of LB. (Kanamycin, 100μg/ml; rifampicin, 50 μg/ml), 28 , 170 rpm, cultured for 12 h, until the bacterial solution OD600 reached to 0.6. Then, the liquid culture was centrifuged at 4000rpm for 10 minutes to collect bacterial cells, resuspended with an appropriate volume of resus- pension solution (10mM MgCl2, 10mM MES, and 200 uM acetosyringone) to a final concentration of  2.0 (OD600), and the suspension was placed at 28°C in the dark for 3 hours. Then the TRV1 solution and the solution containing negative control (TRV::00), positive control (TRV::CLA) (Cloroplastos alterados) and silenced gene (TRV::GhCBP60B) were suspended at a ratio of 1:1 (v/v). After mixing, the seven days old cotton cotyledons were inoculated. Plants obtained after gene silencing were cultured at 23°C with a 16h/8h light/dark cycle. Two weeks after the  inoculation with bacterial solution, the inoculated cotton plants were treated with 20 % PEG6000 and 200 mM NaCl solution for two days. The ultrapure water was used as a control. Three biological replicates and three technical replicates were set up, with at least ten strains in each experimental group. The GhCBP60B gene-specific primer sequences of VIGS are enlisted in Table S4.

3. Line 392- should be:  Li et al. (2021)

4. Line 39, 482- lack of space  

5. Check if the contents of all Supplementary files is consistent with their numbers and description in text.

Comments on the Quality of English Language

Minor editing of English language required.

Author Response

请参阅附件。

Reviewer 3 Report

Comments and Suggestions for Authors

I would like to thank authors for the deep analyses of the comments and questions and full answers. Authors have carefully answered all the questions addressed to the manuscript. All mistakes were corrected, manuscript was reviewed and improved. All missing data were included to the manuscript. Again, some species names still need to be written in Italic type (Line 541). Please check the entire text again for Italic type for the species names.

Author Response

    On behalf of all the contributing authors, I would like to express our sincere appreciations of your careful review and constructive Suggestions ’ constructive comments concerning our article entitled “Genome-wide identification of calmodulin-binding protein 60 gene family and the function of GhCBP60B in cotton growth and development and abiotic stress response” (Manuscript : ijms-2929791). Based on your new suggestions,We have double-checked the italics in the full text. Thank you again for your review of our manuscript, it is our pleasure.
